# A Rotational Invariant Neural Network for Electrical Impedance Tomography Imaging without Reference Voltage: RF-REIM-NET

**DOI:** 10.3390/diagnostics12040777

**Published:** 2022-03-22

**Authors:** Jöran Rixen, Benedikt Eliasson, Benjamin Hentze, Thomas Muders, Christian Putensen, Steffen Leonhardt, Chuong Ngo

**Affiliations:** 1Helmholtz Institute for Biomedical Engineering, RWTH Aachen University, 52074 Aachen, Germany; benedikt.eliasson@rwth-aachen.de (B.E.); hentze@hia.rwth-aachen.de (B.H.); leonhardt@hia.rwth-aachen.de (S.L.); ngo@hia.rwth-aachen.de (C.N.); 2Department of Anaesthesiology and Intensive Care Medicine, University of Bonn, Venusberg-Campus 1, 53127 Bonn, Germany; thomas.muders@ukbonn.de (T.M.); christian.putensen@ukbonn.de (C.P.)

**Keywords:** artificial intelligence, deep learning, Electrical Impedance Tomography, lung imaging, cardiopulmonary monitoring

## Abstract

*Background*: Electrical Impedance Tomography (EIT) is a radiation-free technique for image reconstruction. However, as the inverse problem of EIT is non-linear and ill-posed, the reconstruction of sharp conductivity images poses a major problem. With the emergence of artificial neural networks (ANN), their application in EIT has recently gained interest. *Methodology*: We propose an ANN that can solve the inverse problem without the presence of a reference voltage. At the end of the ANN, we reused the dense layers multiple times, considering that the EIT exhibits rotational symmetries in a circular domain. To avoid bias in training data, the conductivity range used in the simulations was greater than expected in measurements. We also propose a new method that creates new data samples from existing training data. *Results*: We show that our ANN is more robust with respect to noise compared with the analytical Gauss–Newton approach. The reconstruction results for EIT phantom tank measurements are also clearer, as ringing artefacts are less pronounced. To evaluate the performance of the ANN under real-world conditions, we perform reconstructions on an experimental pig study with computed tomography for comparison. *Conclusions*: Our proposed ANN can reconstruct EIT images without the need of a reference voltage.

## 1. Introduction

Electrical Impedance Tomography (EIT) enables the non-invasive visualization of the dielectric properties of a medium of interest. EIT has a wide range of applications, including the status monitoring of concrete [1], the monitoring of semiconductor manufacturing [2], and observing cell cultures [3]. In the medical domain, the applications are broader, and include the monitoring of lung recruitment and collapse [4], lung ventilation [5] and perfusion, the monitoring of 3D brain activity [6], size and volume estimation of the bladder [7], breast cancer imaging [8], and cardiopulmonary monitoring [9]. Here, EIT can be used to assess metrics such as regional ventilation, end-expiratory lung volume, compliance, regional respiratory system compliance, and regional pressure–volume curves [9].

The versatility of EIT stems from the fact that the measurements can be made non-invasively and inexpensively. For an image to be reconstructed, electrodes need to be placed around the domain. Small, low-frequency currents in the range of 100 kHz are fed through these electrodes. Then, the voltage across the electrodes is measured, and an image is reconstructed. Despite the advantages of EIT, it has one major drawback: it suffers from a relatively low spatial resolution.

This issue is due to the fact that EIT image reconstruction belongs to the class of inverse problems [10]. Large changes in the conductivity of the medium may lead to only small changes in the voltage measurements. To still be able to solve the problem, different types of algorithms have been proposed in the literature. From a mathematical perspective, three different types of algorithms can be distinguished.

The first set of algorithms is variational regularization methods. Their goal is to minimize a cost function that contains two parts. First, the physical behavior of the medium of interest is modeled. Given a set of voltage measurements, the algorithm helps to find the best fit for the conductivities that could produce these voltages. Second, the regularization strategy is applied, which plays a crucial role in finding a valid solution. Two common examples of regularization strategies are total variation [11] and the Tikhonov regularization [12].

The second type of algorithms is statistical inversion methods. Here, image reconstruction is modeled as a problem of statistical inference. The measurements and conductivities are modeled as random variables from which an a posteriori distribution can be estimated, through, e.g., Markov Chain Monte Carlo iterations. From this, the conductivity can be derived [13]. This can be accomplished, by, for example, first obtaining a starting distribution through the one-step Gauss–Newton method. Thereafter, Markov Chain Monte Carlo methods can be used to refine the starting distribution [14].

The final type is direct inversion algorithms. In these methods, the problem is analyzed through the partial differential equations governing the system behavior. From this, a solution strategy is developed. An example of these kinds of methods is the D-Bar algorithm [15].

Artificial neural networks belong to the variational regularization methods, as they solve the optimization problem once during training and then act like a complex look-up table. The regularization performed by artificial neural networks is not straightforward: first, the neural network architecture provides a part of the regularization. A very deep architecture may provide sophisticated results for the training data set, but may lead to profound over fitting, such that the results for slightly different data bring far worse results. The second part of the regularization comes from the training data. There is no reference technique to capture the conductivity distribution of body tissue. Thus, in EIT the training data are simulated with the help of, for example, finite element method (FEM) software such as EIDORS [16]. However, for simulations a multitude of assumptions have to be made: What does the model shape look like? What are the electrode positions? Do they change? What shape do the conductivity enclosures have? What is the range of conductivity? All of these assumptions act as some kind of regularization.

Artificial neural networks are beginning to gain more relevance in the field of EIT. In 2017, Kłosowski and Rymarczyk [17] presented an ANN with fully connected layers and convolutional layers. However, the proposed ANN can only reconstruct single targets. Their outputs are the coordinates and the radius of the conductivity enclosure. Other approaches used ANNs to enhance the reconstructions of traditional EIT reconstructions [18]. In 2019, Hu et al. [19] used the spatial invariant properties of the EIT to improve upon these results. However, to aid in the reconstruction, their approach is based on calibration. Thus, their artificial neural network is not usable when the background data are missing. By contrast, Chan et al. [20] proposed a network which does not need this preprocessing. However, the structure of the artificial neural network does not account for the symmetry of EIT measurements. We settled for artificial neural networks, as they have been used in the past within the domain of EIT and show the greatest potential due to their ability to recreate non-linear functions.

In the following, we propose an artificial neural network structure which can reconstruct images without dependence on a reference voltage, while still using the rotational symmetry of EIT adjacent measurements in adjacent drive. We call this structure the **R**eference **F**ree **R**otational **E**lectrical **I**mpedance **M**ap **Net**work (RF-REIM-NET).

The novelties of this research are:We use real-world animal trial data with CT references to confirm that RF-REIM-NET gives meaningful results in such a setting.Our training data are unbiased, as we used a conductivity range bigger than what is expected in the thorax region and did not try to model the conductivity distributions typically encountered in the thorax region.We present a method for time-effective data augmentation using the existing training data.Even though RF-REIM-NET uses fully connected layers, it still preserves the rotational invariance of adjacent measurements.

## 2. Materials and Methods

### 2.1. Fundamentals

In EIT, the goal is to find an optimal conductivity distribution given a set of voltage measurements. When using variational reconstruction methods, this is expressed as
(1)$$\sigma_{rec}=arg min\frac{1}{2}{∥F\left(\sigma \right)-V_{meas}∥}^{2}+\lambda {∥L\sigma ∥}^{2},$$
where σ is the conductivity, F(σ) is the forward model, $V_{meas}$ is the measured voltage, λ is the weighting of the regularization term and L is the regularization matrix. When using artificial neural networks (ANNs), the general scheme of this minimization is still true; however, it is achieved differently. While variational reconstruction methods minimize each measurement according to Equation (Equation 1), ANNs will perform the minimization on a given dataset. This can be formulated as
(2)$$arg min\frac{1}{2}{∥Y^{\prime }\left(V_{meas}\right)-Y∥}^{2},$$
where Y denotes the ground truth value and $Y^{\prime }\left(V_{meas}\right)$ denotes the ANNs output depending on the input of the network. During runtime, the ANN behaves deterministically like a look-up table. When assuming that the dataset represents real measurements, the ANN still minimizes the actual measurements.

### 2.2. Electrical Impedance Maps

Hu et al., pointed out the advantages of packing EIT measurements into the electrical impedance map (EIM). EIMs can be used to represent EIT data in adjacent–adjacent measurement mode. For 16 electrodes, the data are represented in a 16×16 matrix; see Figure 1. Along the matrix column are the measurement electrodes, while the excitation electrodes are arranged along the rows. EIM[j,k] contains the measurement of the *j*th electrode pair, while the *k*th electrode pair drives the current. Since four probe measurements are used, voltages from injecting electrodes cannot be used. On those spots, the EIM matrix is filled with zeros, causing the superdiagonal, diagonal and subdiagonal elements to become 0.

When a conductivity distribution is rotated by an angle of $\frac{2\pi k}{16}$, where *k* is an integer number, the features of the EIM map do not change. The features are moved diagonally across the image. Thus, a convolutional ANN can extract features from the EIM independent of rotation.

### 2.3. Training Data Set

In machine learning, the data set used for training is an important part of the algorithm’s performance [21]. For EIT, there is no general high-resolution ground truth dataset. Instead, the data have to be carefully designed. It is easy and tempting to craft a data set that gives meaningful results on the available test data. If there is a relatively narrow band of possible conductivities in the test data, using this conductivity band in the simulated training data would bring a bias to the network—it might look better than it actually is.

To avoid this fallacy, we designed our data with as few assumptions as possible. We used FEM simulations to create the training data. These simulations were executed using EIDORS [16]. The first practical constraint faced was simulation time, and in general, higher mesh density is better for the quality of the simulations. However, the time taken for meshing and actual computation increases non-linearly. Thus, we used a mesh density of 0.075, while the model radius was chosen to be 28, as this is a feasible trade-off between simulation quality and computation time. Our domain shape was cylindrical. We used 16 electrodes, each with a height of 40 mm and a width of 20 mm. The electrodes were placed equidistantly around the domain. This setup was chosen as it imitates the typical measurement configuration of the clinically available device for thoracic images from Draeger (*Draeger Pulmo Vista 500*, Draeger Medical GmbH, Lübeck, Germany).

#### 2.3.1. Basic Object Shapes

To create conductivity enclosures, we used three different basic shapes: an ellipsoid, a cube and an octahedron. The basic size of these objects is 1 in all directions, and their center of gravity is in the origin of the coordinate system. To save computation time, we did not re-mesh each impedance enclosure from scratch. Instead, we created a mask for each conductivity enclosure and then changed the conductivities of mesh elements inside the mask *m*. The formulas for the three basic shapes are given as:(3)$$mask_{\mathrm{sphere}}=\left(x^{2}+y^{2}+z^{2}\right)\leq 1$$
(4)$$mask_{\mathrm{cube}}=\max\left(|x|,|y|,|z|\right)\leq 1$$
(5)$$mask_{\mathrm{octahedron}}=\left|x\right|+\left|y\right|+\left|z\right|\leq 1$$

#### 2.3.2. Transformation of the Basic Objects

Only inserting the same shape at the same place in the FEM model would be of no use for real-world reconstructions. Thus, the basic shapes have to be transformed. Our transformation involves the translation, rotation and scaling of the enclosures. This can be mathematically described as:(6)$$v^{\prime }=\left(R\left(v-t\right)\right)⊘s,$$
where v={x,y,z} is the coordinate vector, $v^{\prime }$ is the transformed coordinate vector, $t\in R^{3}$ is the translation vector, $R\in R^{3\times 3}$ is the rotation matrix, $s\in R^{3}$ is the scaling vector and ⊘ denotes the element-wise division.

The positioning of the enclosures is important. As the ANN should be able to detect any conductivity enclosures in the domain with the same quality, the distribution of the object’s center of gravity should be uniform across the domain. Thus, we sampled the values of t from a uniform distribution, such that every component of t={x,y,z} is well inside the domain boundaries. Figure 2 gives a visual example of the transformations applied to the data.

Each entry of the scaling vector s is uniformly sampled between 10% and 80% of the model radius.

The angle of the rotation matrix R is uniformly sampled from [0,2π), and thus the basic shape can be rotated in any direction. This enables the ANN to learn features that are valid for a variety of positions, as only the position of the feature changes. In Figure 2, the transformation is visualized.

#### 2.3.3. Conductivity Range

Another important degree of freedom is the conductivity range used. When using EIT tanks for testing, the conductivity range of the test data is typically known. Thus, it would be very tempting to just use this conductivity range for the training of the ANN. However, in practice, the conductivity is typically not known to this level of detail. Through a slice of the chest, conductivity values can range from $3.5\times 10^{-3}$ S/m (cortical bone) up to $4.64\times 10^{-1}$ S/m (deflated lung) [22]. We used a range of $1\times 10^{-5}$ S/m to 1 S/m for the background conductivity, as this covers the conductivity values typically encountered in chest measurements, while at the same time providing a margin well outside to improve generalization. The values were sampled uniformly from a logarithmic arrangement of the mentioned conductivity range. This is also known as a reciprocal or log-uniform distribution. This was chosen because the ANN should be able to differentiate objects that are one order of magnitude bigger than the background, regardless of the actual background conductivity.

The next step is an appropriate choice of the conductivity enclosures. As mentioned, it is important that the ANN is able to distinguish conductivity contrasts. At the same time, the ANN shall also be able to distinguish those contrasts symmetrically in the lower and upper bound of the conductivity range. To achieve this, the enclosure’s conductivity is chosen with respect to the background conductivity. This ensures that the ANN has no bias towards a conductivity contrast higher or lower than the background. Thus, the enclosure conductivity is chosen by multiplying the background conductivity with values from a range of $1\times 10^{-2}$ to $1\times 10^{2}$. Again, this is sampled uniformly from the logarithmic arrangement of those values.

In a real-world setting, conductivities are rarely perfectly homogeneous across a tissue type. Because of this, the enclosures, as well as the background, are perturbed. We again scale each node of the FEM model by different values. This is achieved by using a Gaussian distribution with a *mean* value of 1. For each training sample and chosen conductivity value, a different standard deviation (*std*) from $1\times 10^{-8}$ to $1\times 10^{-2}$ was chosen. When the *std* is chosen, the values of one conductivity are perturbed by multiplication with the sampled values.

#### 2.3.4. Electrode Contact Impedance

Another effect to consider is the electrode contact impedance. Although the adjacent drive pattern used here relies solely on four probe measurements and, thus, will reduce the effect of electrode contact impedance, we included the effect into our training data. We multiplied EIDORS default contact impedance by a value randomly sampled from a Gaussian distribution with a *mean* value of 1. To simulate high and low differences, we sampled the values from three different distributions with an *std* of $1\times 10^{-5}$, $1\times 10^{-3}$ and $1\times 10^{-1}$.

#### 2.3.5. Measurement Noise

ANNs typically struggle with generalizing learned samples to cases that the ANN has not yet seen [23]. To tackle this problem, further data augmentation strategies need to be used. While the previously mentioned steps required new simulations for each training sample, the following steps rely on already simulated data. This saves computation time.

EIT measurements can be affected by several sources of noise. Paired with the ill-conditioned nature of the EIT problem, this can cause artifacts in the reconstruction. Often, reconstruction algorithms have a hyperparameter, which in essence balances the robustness to noise and the quality of the reconstruction. As for ANNs, the sensitivity to noise can be adjusted through the noise in the training data.

A major component in EIT systems is the analog digital converter; the noise consists primarily of thermal, jitter, and quantization noise [24]. The first two depend on the magnitude of the signals. The greater the signal, the bigger the noise. We can model this by multiplying the noise-free signal with a constant drawn from Gaussian noise:(7)$$U_{\mathrm{ther},\mathrm{jit}}^{i,j}=U^{i,j}\;\cdot \;n_{\mathrm{mult}},\;\;n_{\mathrm{mult}}∼N\left(1,\sigma^{2}\right),\forall i,j\in \left\{1,\ldots ,16\right\}$$
where $U_{\mathrm{ther},\mathrm{jit}}^{i,j}$ is the thermal noise-affected measurement, $U^{i,j}$ is the noise-free measurement and $n_{\mathrm{mult}}$ is the noise sampled from a normal distribution. The quantization noise does not depend on the signal level, and can be modeled by adding a noise term to the voltage signals:(8)$$U_{\mathrm{quant}}^{i,j}=U^{i,j}\;+\;n_{\mathrm{add}},\;\;n_{\mathrm{add}}∼N\left(0,\sigma^{2}\right),\forall i,j\in \left\{1,\ldots ,16\right\}$$
where $U_{\mathrm{quant}}^{i,j}$ is the quantization noise-affected measurement and $n_{\mathrm{add}}$ is the noise sampled from a normal distribution. However, there is still another source of noise. Different measurement channels of a given EIT system can have different gains. This is due to different gains in the multiplexers [25]. The noise can be described through
(9)$$U_{\mathrm{gain}}^{i,j}=U^{i,j}\;\cdot \;n_{\mathrm{mult}},\;\;n_{\mathrm{mult}}∼N\left(0,\sigma^{2}\right),\forall j\in \left\{1,\ldots ,16\right\}$$
Note that this noise only affects one row of the EIM, compared with Equation (Equation 7), where every entry is affected individually. For the additive noise, $n_{\mathrm{add}}$ a *std* of $1\times 10^{-8}$ was chosen. For the multiplicative noise, $n_{\mathrm{mult}}$, an *std* of $1\times 10^{-6}$ was chosen.

#### 2.3.6. Rotation of the Data

To specifically incorporate the rotational invariance into the ANN, the voltage data were prepared with minimal computational costs, as follows. A shift of *n* columns along the EIM results in a rotation of the reconstructed image by $\frac{2\pi n}{16}$. Thus, the target image must be shifted according to that angle. To get rid of the rotational variance, we produced 15 additional shifted voltages for each training sample, described previously.

#### 2.3.7. Alpha-Blending

With the given data set, there is still potential for obtaining entirely new training samples. In the field of image classification, there is a technique called α-blending [26,27,28]. It produces a new image from a linear combination of two other images, and an α∈[0,1] factor weights these images. An α of 0.3 would mean that the resulting image is a combination of 30% of the first image and 70% of the second image. For EIT images, we can describe the technique as
(10)$$\sigma_{\mathrm{comb}}=\alpha \sigma_{1}+\left(1-\alpha \right)\sigma_{2}$$

From Ohm’s law with conductivities and the constant injection current follows the procedure to combine the voltages accordingly
(11)$$Y_{\mathrm{comb}}=\alpha Y_{1}+\left(1-\alpha \right)Y_{2}$$
(12)$$\Leftrightarrow \frac{I}{U_{\mathrm{comb}}}=\alpha \frac{I}{U_{1}}+\left(1-\alpha \right)\frac{I}{U_{2}}$$
(13)$$\Leftrightarrow \frac{1}{U_{\mathrm{comb}}}=\alpha \frac{1}{U_{1}}+\left(1-\alpha \right)\frac{1}{U_{2}}$$
(14)$$\Leftrightarrow U_{\mathrm{comb}}={\left(\frac{\alpha }{U_{1}}+\frac{1-\alpha }{U_{2}}\right)}^{-1}$$
where Y denotes the admittance between the voltage measurement electrodes.

#### 2.3.8. Conclusion on Trainign Dataset

All in all, the choice of the simulated training data was made such that it was as realistic as possible, but at the same time no major assumptions were made on the structure and content of the simulated data nor on the bias in the dataset (e.g., restricting conductivity values to the range expected in the testing data). Furthermore, the described augmentation techniques impose no bias on the simulated data.

### 2.4. On the ANN Structure

In the domain of classification, ANNs can often be separated into two parts. The first part, consisting of convolutional layers, is used for the extraction of features, while the second part is used for processing these features into educated guesses about the class label. Two very famous examples are AlexNet and VGG19 [29,30]. The second part is realized through fully connected layers. In this work, we modified this basic approach and tailored it specifically for use in EIT. The structure can be seen in Figure 3. When calculating the receptive field of the convolutional layers, it can be seen that the receptive field is of the shape 21 × 21, although the EIM only has a shape of 16 × 16. However, Luo et al. showed that the receptive field exhibits a Gaussian distribution [31], which means that features in the center are strongly recognized by the network, while closer to the boundary the features are less recognized. To dampen this problem, we increased the receptive field of RF-REIM-NET.

From Figure 3, it can also be seen that the shape of the input after the convolutions does not change. To achieve this, we used circular padding rather than the standard zero padding. This choice can be understood by the nature of adjacent–adjacent measurements. On the boundaries of the EIM, values from the other side are inserted, as the neighbor of the 16th electrode pair would also be the 1st electrode pair.

Our proposed RF-REIM-NET structure also comes without any form of pooling layers. In general, pooling layers tend to increase the efficiency of ANNs; however, this comes at the cost of broken location invariant properties of the convolutional layers [32]. Another problem is that down sampling, when carried out by pooling, causes aliasing [33]. Thus, we did not use pooling and increased the receptive field of the RF-REIM-NET.

Instead of batch normalization, we used layer normalization. Instead of normalization along the batch, layer normalization computes the normalization along the features of the layer’s output. We found that this works best for training.

The second part of RF-REIM-NET consists of a fully connected layer, adapted such that the rotational invariance is considered. The input to the first fully connected layer has the shape 16 × 2048. This was purposeful, as the 16 represents the 16 different current injection pairs. Thus, instead of passing a vector of 16 × 2048 = 32,768, 16 passes of a 2048 vector are used. This saves time during training and considerably decreases the size of the RF-REIM-NET. The second fully connected layer works the same way, but at the same time the output will be doubled to obtain a 64 × 64 reconstruction image.

### 2.5. Training of the Neural Network

Our training procedure was as follows. α-blending was used during training, and two batches were randomly combined as described in the methods section. As a regularization strategy, L2 weight regularization, dropout with a dropout rate of 0.1 and total variation (TV) regularization were used. L2 weight regularization was added, scaled to the loss of RF-REIM-NET, and can be described as
(15)$$L_{w^{2}}=\sum_{i=1}^{N}w_{i},$$
where *N* is the total number of weights and $w_{i}$ is the $i^{\mathrm{th}}$ weight. For details about dropout, see [34]. TV regularization is commonly used for image de-noising and de-blurring [35]. The TV regularization loss is computed with
(16)$$L_{TV}=\sum_{i,j}|Y_{i+1,j}^{\prime }-Y_{i,j}^{\prime }\left|+\right|Y_{i,j+1}^{\prime }-Y_{i,j}^{\prime }|,$$
where $Y^{\prime }$ is the output of RF-REIM-NET.

For the loss function we used the mean squared logarithmic error (MSLE). This error was chosen as our training data vary in orders of magnitude, and RF-REIM-NET should be able to predict the conductivities in the same way across the whole range. MSLE can be described as
(17)$$L_{MSLE}=\frac{1}{N}\sum {\left(\log\left(Y+1\right)-\log\left(Y^{\prime }+1\right)\right)}^{2},$$
Y is the ground truth conductivity distribution.

The total loss of RF-REIM-NET is described by
(18)$$L_{total}=L_{w^{2}}+\lambda_{1}\;\cdot \;L_{TV}+\lambda_{2}\;\cdot \;L_{MSLE}$$
with $\lambda_{1}=0.1$ and $\lambda_{2}=1\times 10^{-6}$.

RF-REIM-NET was then trained with the help of the TensorFlow [36] and the Adam optimizer [37]. Additionally, we used a learning rate decay: whenever the loss reached a plateau, the learning rate was reduced by 70%.

### 2.6. Evaluating RF-REIM-NET

We compared RF-REIM-NET to the standard Gauss–Newton (GN) reconstruction for absolute EIT. To compare the two algorithms quantitatively, we used a modified version of the GREIT figures of merit [38]. Some modifications were necessary, as the original GREIT figures of merit only allow the evaluation of difference images.

First, the calculation of the evaluation mask needs to be modified. The median value of the reconstructed image is subtracted from the original image,
(19)$$\sigma_{eval}=\sigma -\tilde{\sigma },$$
where $\tilde{\sigma }$ denotes the median value of σ. The median was chosen, as it is more robust to outliers in the data. When inserting only one target for evaluation, an ideal reconstruction would have just two values, the background and the target. If we then subtracted the *mean*, the background would be slightly negative. This is prevented with the help of the subtraction of the median. The mask is then composed of all values, which are 50% less or more than the minimum/maximum value. The choice is dependent on the value of the target with respect to the background. In our evaluation case, the target is less conductive than the background. Thus, our mask is defined as
(20)$$\begin{array}{c} m=\left\{\begin{array}{c} 1\;if\;\sigma_{eval}<\frac{1}{2}\;\cdot \;\min\left(\sigma_{eval}\right)\\ 0\;else \end{array}\right., \end{array}$$
where m is the evaluation mask. We denote all pixels inside the mask as ${\hat{\sigma }}_{eval}$, while all pixels outside the mask are denoted as $∼{\hat{\sigma }}_{eval}$.

#### 2.6.1. Amplitude Response (AR)

The AR is now defined as
(21)$$AR=\sum {\hat{\sigma }}_{eval}.$$
The *std* of the AR should be low.

#### 2.6.2. Position Error (PE)

The PE is defined as

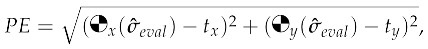
(22)
where 
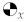
 denotes the x-component of the center of gravity, 
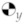
 denotes the y-component accordingly, $t_{x}$ is the ground truth x-position and $t_{y}$ the ground truth y-position. The *mean* and the *std* of the PE should be low.

#### 2.6.3. Ringing (RNG)

The RNG was defined as the *std* of all pixels outside the mask m. Formally, this is written as
(23)$$RNG=std(∼{\hat{\sigma }}_{eval}).$$
The *mean* and standard RNG deviation should be low.

### 2.7. Evaluation Data

To validate our RF-REIM-NET, we used three different types of input and analyzed the output according to the three introduced figures of merit (AR, PE and RNG).

#### 2.7.1. FEM Data

First, we used FEM data that the network had not yet seen. Multiple enclosures were simulated, and the enclosure was positioned such that it move from the domain center to the outside. RF-REIM-NET is compared with GN. For the hyperparameter selection of GN, we at first used the L-curve criterion, but did not find usable results. We are convinced that this is due to the reference-free reconstruction, which destabilizes the EIT problem compared with differential EIT. Thus, we made multiple sweeps of the hyperparameter to narrow down the optimal hyperparameter iteratively.

#### 2.7.2. Noise Performance on FEM Data

Second, we compared the noise performance on a FEM data sample. For that, an enclosure near the boundary was simulated and the noise level was increased from 200 to 5 db. For evaluation, GN reconstructions are given.

#### 2.7.3. Tank Data

Third, we used data from a circular EIT tank. The tank had a diameter of 28 cm and had 16 electrodes attached equidistantly around the surface. The tank was filled with 0.9% saline solution and the target was a pickle with a circumference of 4.5 cm. The pickle was moved from the center in the direction of one electrode in nine steps, where the last position was 9 cm in front of an electrode. The measurements were performed with the EIT evaluation kit 2 (*Draeger EEK2*, Draeger Medical GmbH).

#### 2.7.4. Experimental Data

Finally, we give an impression of the performance of RF-REIM-NET on real-world data. The data were taken from an experimental pig trial using a clinical EIT device (*Draeger Pulmo Vista 500*, Draeger Medical GmbH). For the trial, eight pigs were anesthetized and tracheotomized in supine position [39]. During the trial, CT measurements from the pigs were taken. In our data sample, we used two measurement points from a single pig, which was healthy in the time span we chose. The length of the data sample was around 30 s.

As there is no ground truth regarding the conductivity, we show two pictures. The first picture shows the *mean* conductivity over an entire breathing cycle, while the second shows the *std* over an entire breathing cycle. As the background, a CT image is given, as it will give a sense of quality. This is given for transparency, as tank data have more ideal conditions, which are closer to the training data. The absolute Gauss–Newton algorithm did not yield any meaningful results after a thorough hyperparameter sweep and was thus not given as a reference.

## 3. Results

At first, we give the results of simulated FEM data, outside the distribution of the training data used to train RF-REIM-NET. The results for the figures of merit are given in Table 1. The *std* of the AR is bigger for the GN algorithm; however, the *std* is one order of magnitude lower compared with the *mean*. Visually, this is confirmed by the reconstructions in Figure 4. The AR for both reconstructions stays roughly the same. For the PE, GN has a lower *mean* and *std*. The PE *mean* is half that of the RF-REIM-NET, while the *std* is a quarter. The RNG for the GN is also lower compared with RF-REIM-NET. This, again, can be seen in the images. The RNG for RF-REIM-NET is larger, due to the higher differences in magnitude outside the mask m.

To better see the difference between the original ground truth image and the reconstruction, we present in Figure 5 the ground truth, the reconstruction of RF-REIM-NET and the MSLE error. It can be seen that the error is for the most part on the edges of the enclosures. In the second column, we can see that the middle target is barely visible in the reconstruction.

### 3.1. Noise Comparison on Simulated Data

Here, we compare the noise performance of the RF-REIM-NET compared with GN. We positioned a target near the boundary of the FEM domain and simulated the voltages. In Figure 6, the results can be seen. The reconstructions of RF-REIM-NET are more robust to noise, compared with GN. At 100 db the reconstruction of GN is barely visible, while RF-REIM-NET is still clearly visible. At 15 db the reconstruction of RF-REIM-NET begins to degrade and also becomes less visible.

### 3.2. Tank Results

Next, we provide the results from the tank measurement. As shown in Figure 7, the pickle was moved from the center to an electrode. The reconstruction from the Gauss–Newton algorithm shows a more diffuse boundary, while RF-REIM-NET has a more clear boundary. The background from the Gauss–Newton algorithm shows many, but small background disturbances, while the background of RF-REIM-NET has fewer disturbances, where one is at the top and the other surrounds the conductivity enclosure.

These observations are also reflected in the figures of merit given in Table 2. The *mean* and *std* of the AR from RF-REIM-NET are bigger than the ones of Gauss–Newton. This can also be visually confirmed in Figure 7. The PE and its *std*, however, are lower in RF-REIM-NET. The *mean* of the PE from RF-REIM-NET is ∼35% lower than that of Gauss–Newton, while its *std* of the PE is ∼50% lower. The *mean* RNG of RF-REIM-NET is 20% lower compared with Gauss–Newton. However, the inverse is true for the *std*: the RNG *std* is 20% higher compared with Gauss–Newton. However, the *std* is $\frac{1}{14}$th of the *mean*.

### 3.3. Experimental Data

For the experimental data, only qualitative analysis is given. The results are shown in Figure 8. On the left, a CT measurement is given to better judge the results. In the middle, the mean reconstruction over 20 s of mechanical ventilation is given. At the top, there is an artifact in the reconstruction. The two lungs are visible, but they are smaller compared with the original size. The heart on the other hand, between the lungs and the artifact, matches the position given in the CT image. The standard deviation picture on the right confirms the findings. The artifact stays in the position, as the standard deviation is near zero at this position. At the position of the heart a high standard deviation is visible, and the same holds true for the lungs. The shape of the standard deviation picture for the lungs better resembles the general shape of the lung.

### 3.4. Discussion

In the FEM setting, GN outperformed RF-REIM-NET in the metrics of PE and RNG, both with the mean and the *std*. The mean and *std* of the AR are a little higher using GN. However, the values only differ slightly. Thus, we would argue that GN outperformed RF-REIM-NET in the FEM setting. This is probably due to the fact that the setting has not much disturbance by factors such as hardware or the imperfect conductivity of the target.

In a tank setting, RF-REIM-NET has a lower mean PE and mean RNG. The PE also has a *std* that is roughly half that of the GN PE *std*. This can be visually observed in Figure 7. However, at the same time, the mean AR and its *std* is higher. This, again, can be seen in Figure 7. In the samples “pickle 2”, “pickle 3” and “pickle 4”, the reconstruction is clearly larger than in the other samples. In contrast, GN has a less clear object boundary. Thus, we argue that on the tank dataset, RF-REIM-NET has a better performance. We showed that RF-REIM-NET is able to give reconstructions from experimental data, even though the ANN does not need any reference voltage, as can be seen in Figure 8. To the best of our knowledge, this is the first work to evaluate the performance of ANNs for EIT reconstructions on real-world experimental data from an ANN solely trained on simulated data.

While the heart was reconstructed accurately, the lungs were too small, which is at that point not fully useful for clinical diagnostics. Another shortcoming is the artifact at the top, which constantly stays in that position. We assume that the artifact is due to electrode position errors. At the top of the picture, the EIT belt is closed. Thus, the electrodes have a larger distance from each other at that place.

## 4. Conclusions and Outlook

We present an ANN (RF-REIM-NET) that is able to reconstruct conductivity enclosures without using a reference voltage. RF-REIM-NET is inspired by ANNs that are commonly used for classification: the first part of these ANNs extracts features, while the second part is responsible for the evaluation. Compared with GN on FEM and tank data, our approach tends to give clearer reconstructions. However, the images tend to be a little bigger than in real life. We also showed the performance on real-world subject data. Compared with GN, which did not obtain any meaningful reconstructions from the experimental data set, RF-REIM-NET was able to give reconstructions. For future work, the network needs to be made more robust against electrode position errors and domain shape influences, which may be the biggest impact factors on the experimental data performance. Thus, in the future, altering the electrode positions in the training data might improve the overall reconstructions. Second, the boundary shape has to be altered more drastically, as this might further increase the performance of RF-REIM-NET.

## Figures and Tables

**Figure 1 diagnostics-12-00777-f001:**
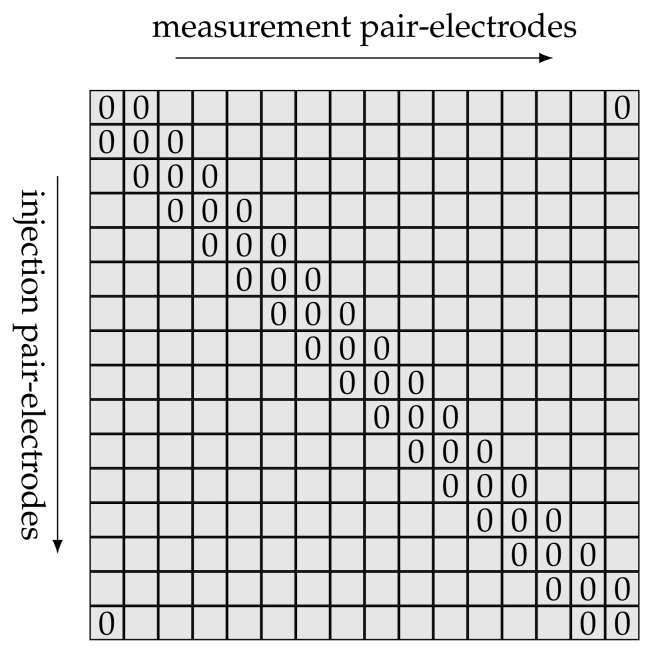
A 16 × 16 electrical impedance map (EIM) arrangement from an adjacent injection pattern. The zeros represent values which are not gathered from adjacent–adjacent measurement mode, as at least one electrode is used for current injection.

**Figure 2 diagnostics-12-00777-f002:**
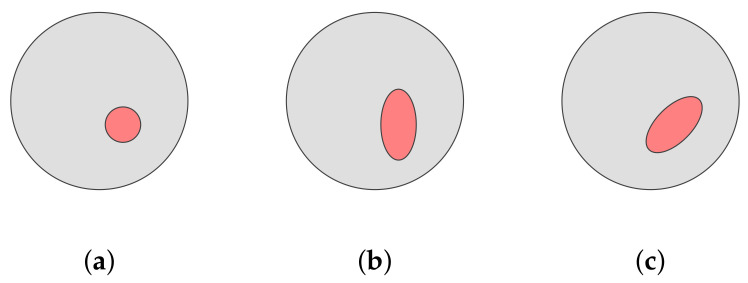
An example of the random transformation used in dataset generation, applied to a circular shape. (**a**) Circle with offset. (**b**) Circle with offset and scaling. (**c**) Circle with offset, scaling and rotation.

**Figure 3 diagnostics-12-00777-f003:**
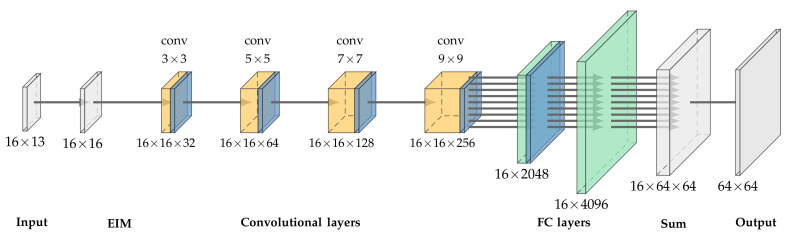
Illustration of the RF-REIM-NET structure. At the beginning, the 16×13=208 voltages are transformed into an EIM. From these, features are extracted with the help of convolutional layers. At the end, these features are processed by fully connected layers, which reconstruct the image for each injection electrode.

**Figure 4 diagnostics-12-00777-f004:**
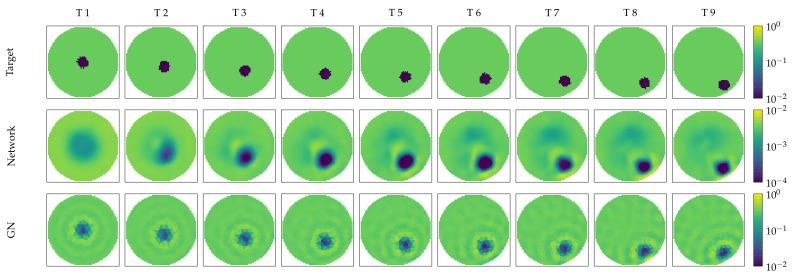
Illustration of RF-REIM-NET (**middle**) and GN (**bottom**) reconstructions of FEM that mimic the position of the tank pickle data. The ground truth target positions are given at the (**top**).

**Figure 5 diagnostics-12-00777-f005:**
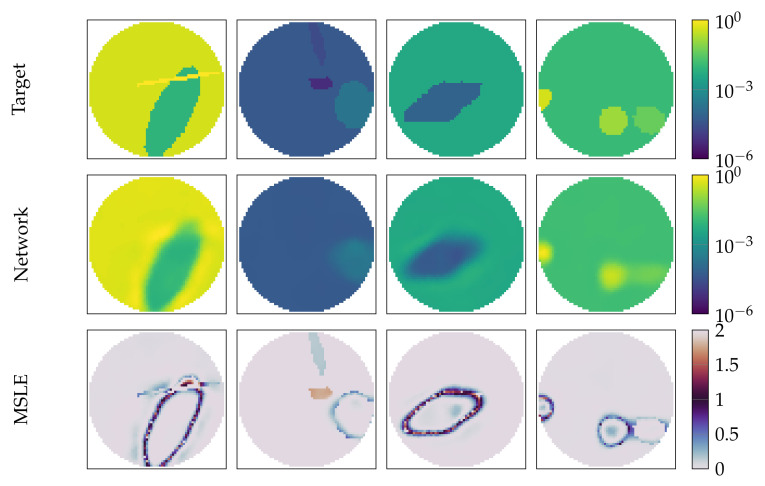
Illustration of RF-REIM-NET reconstructions on the validation dataset. At the top, the original targets are given. In the middle, the reconstructions of RF-REIM-NET are presented. At the bottom, the MSLE error between the original image and the reconstruction are presented.

**Figure 6 diagnostics-12-00777-f006:**
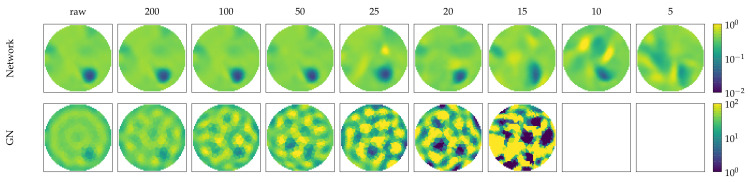
Evaluation of the noise performance of RF-REIM-NET (**top**) compared with GN (**bottom**). The columns represent different noise levels added with the EIDORS function add_noise. The target position is equal to T6 in Figure 4.

**Figure 7 diagnostics-12-00777-f007:**
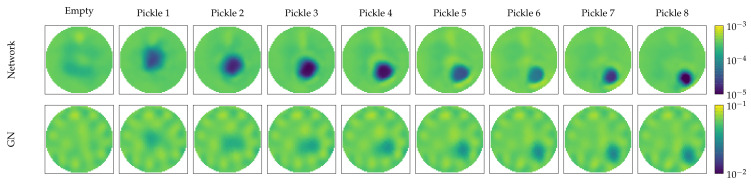
Illustration of RF-REIM-NET and GN reconstructions on the pickle measurements of the tank dataset. The empty measurement was not used in both reconstructions, and is given only for a better view of the reconstruction artifacts. Pickle 1 is the center pickle, while Pickle 7 is the outer pickle.

**Figure 8 diagnostics-12-00777-f008:**
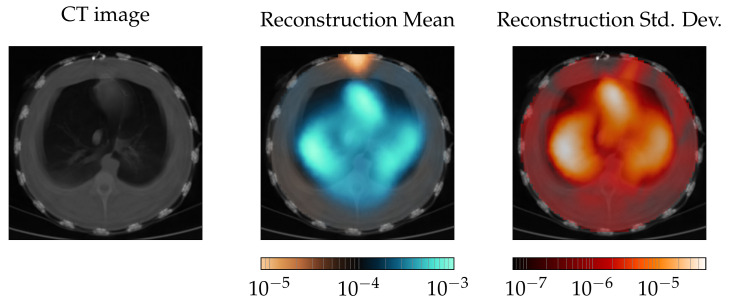
Comparison of the RF-REIM-NET reconstruction with CT scans. On the very left, the CT scan of the pig thorax is given. In the middle, the mean over 20 s of mechanical ventilation is given. On the very right the *std* is given.

**Table 1 diagnostics-12-00777-t001:** Figures of merit for the simulated FEM data. Given is the *mean* ± *std*.

Algorithm/Metric	AR	PE	RNG
GN	0.069±0.0069	2.7±1.1	0.11±0.0094
RF-REIM-NET	0.066±0.0046	5.6±4.0	0.14±0.019

**Table 2 diagnostics-12-00777-t002:** Figures of merit for the tank experiment. The left number in each cell is the *mean* of the metric, while the right number is its *std*.

Algorithm/Metric	AR	PE	RNG
GN	0.1±0.0056	11±5.7	0.1±0.0045
RF-REIM-NET	0.14±0.0086	7.1±2.9	0.08±0.0054

## Data Availability

Not applicable.

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
