# Peer review of "A Rotational Invariant Neural Network for Electrical Impedance Tomography Imaging without Reference Voltage: RF-REIM-NET"

_diagnostics, 2022, doi:10.3390/diagnostics12040777_

Round 1
Reviewer 1 Report
This article is an interesting approach in the aspect of the application of computational intelligence algorithms in electrical tomography. Nevertheless, in methods based on machine learning, an important element is the learning and validation set. The effectiveness of the prediction and the thesis depends largely on the selection of criteria and the amount of data and the appropriate selection of data. The universality of the algorithm thus has some limitations.
Minor remarks:
1) I suggest describing in more detail on what basis and from what the scope of the selection of input data for the research problem posed.
2) The authors could refer to other machine learning methods and justify the choice of the methods presented.
Author Response
Dear reviewer,
thank you for your valuable feedback. You are perfectly right, that machine learning typically requires a split between training dataset and test dataset. As for electrical impedance tomography (EIT), we have this split implicitly. For training, we used a simulated dataset as described. For the testing dataset, we used data collected from a physical EIT tank. No samples from the tank dataset were ever used for training. Thus, a split between the test and training dataset is given. We note that simulations trail behind data collected in the real world, thus the choice of the tank dataset as testing dataset is even a disadvantage for the neural network.
The choice of the simulated training data was made such, that it is as realistic as possible, but at the same time make no major assumptions on the structure and content of the simulated data nor bias the dataset (e.g. the conductivity values simulated were chosen such, that the values contain not only conductivities encountered in pigs, but also beyond that). For future research, it is interesting to evaluate the influence of the choice of training data on the performance of the neural network.
We added a subsection (2.3.8) to the manuscript to explain this in more detail.
As past machine learning results have shown relatively promising results based on neural networks, we have selected these too. Neural networks offer non-linear capabilities, while at the same time promise good control during training (e.g., regularization strategies like drop-out, weight decay). We also added this into the introduction to make this more clear.
All changes are marked in red for better readability.
Best regards, on behalf of all the authors
Reviewer 2 Report
I congratulate the authors for this high quality manuscript on very interesting topic. ANN seem to be promising for absolute EIT imaging.
My main comment would concern comparison with Gauss-Newton. The selection of the hyperparameter is influencing the noise properties and resolution. It is sometimes arbitrarily chosen, or sometime the L-curve method is used, but requires a specific conductivity image to be selected. If I am not mistaken, you did not describe this in detail. Could this be added ?
My last minor comment concerns alpha-blending. The main assumption in Equation (10) would be valid only under certain circumstances, following the theory of the "Rule-of-Mixtures". I admit hearing about the theory for lung applications and never finding time to dig into the topic. I would simply recommend the authors to check if it is worth a comment in the section on alpha-blending.
I recommend already the manuscript for publication, and thank the authors for consideration of my two comments.
Author Response
Dear reviewer,
we thank you very much for your valuable comments. We ran LanguageTool (a professional spell and grammar checking tool like Grammarly) through the script, to double-check for minor errors.
For the comparison with the Gauss Newton algorithm, we found that the choice of the hyperparameter through the L-curve criterion did not yield strong results. As with this hyperparameter, the reconstructions did not look clear on the data. The results were even worse for the real-world tank data. We think this might be due to the fact, that we are using a reference free reconstruction algorithm. These are more unstable in the case of electrical impedance tomography (EIT) compared to differential EIT algorithms. Thus, we made several sweeps of reconstructions with different hyperparameters and judged the reconstruction quality. We did this several times to get closer to the optimal value. In the results, you can see that the Gauss Newton algorithm even outperforms the neural network on the tank data. We added this in section 2.7.1 (changes are highlighted in red).
Thank you, also, for the hint with the “rule of mixtures” theory. We looked further into this theory and come to the conclusion that in our case this theory is not of concern. The theory is for composite materials, in our case we want to simulate with equation 10 further data samples of new types rather than composite materials. However, we are truly thankful for this insight, as this theory is really helpful in another domain of our research. Thank you for that.
Best regards, on behalf of all the authors